# ON TRAINING DERIVATIVE-CONSTRAINED NEURAL NETWORKS

## ABSTRACT

We refer to the setting where the (partial) derivatives of a neural network's (NN's) predictions with respect to its inputs are used as additional training signal as a *derivative-constrained* (DC) NN. This situation is common in physics-informed settings in the natural sciences. We propose an integrated RELU (IReLU) activation function to improve training of DC NNs. We also investigate *denormalization* and *label rescaling* to help stabilize DC training. We evaluate our methods on physics-informed settings including quantum chemistry and Scientific Machine Learning (SciML) tasks. We demonstrate that existing architectures with IReLU activations combined with denormalization/label rescaling better incorporate training signal provided by derivative constraints.

## 1 INTRODUCTION

Deep learning is increasingly being applied to physics-informed settings in the natural sciences. By *physics-informed*, we mean any situation where inputs and/or outputs in a dataset involve relationships based on physics (*e.g.*, forces). The field of Scientific Machine Learning (SciML) (Karniadakis et al., 2021) is emerging to address the issues of applying machine learning (ML) to the physical sciences (*e.g.*, physics-informed neural networks or PINNs (Raissi et al., 2019)). Domains include fluid dynamics (Sun et al., 2020; Sun & Wang, 2020), geo-physics (Zhu et al., 2021), fusion (Mathews et al., 2020), and materials science (Shukla et al., 2020; Lu et al., 2020). In the realm of quantum chemistry, there are promising results (Hermann et al., 2022; 2020) that attempt to solve the electronic-structure problem (i.e., predict energy from structure) or model an atomic system's energy surface (Hu et al., 2021; Gasteiger et al., 2021)

In the physics-informed setting, it is common to express constraints on the neural network's (NN's) predictions in terms of the NN's (partial) derivatives with respect to (w.r.t.) its inputs to express physical constraints. We call this a *derivative-constrained* (DC) NN. Thus, training a DC NN addresses a subset of issues considered in physics-informed settings such as SciML. We emphasize that most settings do not use derivatives of the model w.r.t. its inputs to supply additional training signal even though most NN models are optimized with gradient-based methods.

One strategy for incorporating derivative constraints is to add additional terms containing the derivative constraints to a loss function so that multi-objective optimization can be performed. While it is possible to construct NNs with high predictive accuracy using this strategy, the resulting models may not incorporate derivative constraints efficiently. In the physics-informed setting, this translates into capturing less of the physics. We demonstrate that this occurs in many existing works in quantum chemistry and SciML where we obtain high predictive accuracy but lower accuracy on derivative constraints (see experiments, Sec. 5). This presents an opportunity to reevaluate aspects of training DC NNs and revisit best practices. We make the following contributions.

1. We propose a new activation function called an *integrated ReLU* (IReLU) obtained by integrating a standard ReLU activation (Agarap, 2018) (Sec. 4.1). We intend IReLU's as a drop in replacement for activations in an existing architectures. Our main motivation for doing so is that training a DC NN involves higher-order derivatives. Consequently, the choice of activation function will impact the propagation of additional derivative information.

2. We propose *denomalizing* NNs, *i.e.*, removing all normalization layers, and *label rescaling* as a dataset preprocessing method to stabilize training (Sec. 4.2). Our primary motiva-

tion for doing so is because we hypothesize that DC training of NNs is sensitive to *units*. Consequently, unit-insensitive normalization procedures (*e.g.*, batch normalization (Ioffe & Szegedy, 2015)) that help stabilize training in standard settings may introduce artifacts in the DC training case.

We benchmark the performance of our proposed methods on a variety of datasets and tasks including quantum chemistry NNs (Schütt et al., 2017; Xie & Grossman, 2018; Gasteiger et al., 2020b;a; Hu et al., 2021; Gasteiger et al., 2021) and PINNs (Raissi et al., 2019) (Sec. 5) used in SciML. We show that IReLUs combined with denormalization/label rescaling improve the learning of gradient constraints while retaining predictive accuracy.

## 2  RELATED WORK

There are at least two paths to improving training of DC NNs: (1) improving the loss function and (2) developing new architectures. In the first direction, loss functions used in training DC models often involve multiple terms so they are multi-objective optimization (MOO) problems. One solution to the MOO problem is to weigh each term in the loss function (Sener & Koltun, 2018; van der Meer et al., 2022; Bischof & Kraus, 2021), potentially in an adaptive manner (Li & Feng, 2022; Fernando & Tsokos, 2021; Xiang et al., 2022; Chen et al., 2018; Malkiel & Wolf, 2020; Heydari et al., 2019; Kendall et al., 2018; Lin et al., 2017). This is helpful in the SciML context since the different loss terms may use different units of measurements, and thus, have imbalanced label magnitudes (Wang et al., 2021). Weighing loss terms is also common in training quantum chemistry networks. We will demonstrate in Sec. 3.2 that it is difficult to control to motivate our methods.

In the second direction, we can also develop novel architectures that better incorporate domain knowledge. Domains such as quantum chemistry have custom designed NN architectures (Schütt et al., 2017; Xie & Grossman, 2018; Gasteiger et al., 2020b;a; Hu et al., 2021; Gasteiger et al., 2021; 2022; Schütt et al., 2021; Zitnick et al., 2022; Passaro & Zitnick, 2023; Liao et al., 2023). The architectural improvements in these works focus on re-arranging interaction patterns (*e.g.*, convolution layers), leveraging graph properties of atoms (*e.g.*, molecular bond), and encoding invariances/equivariances. We propose an activation function in Sec. 4.1 which we intend as a drop-in replacement for activations in existing architectures. Thus, we intend our activation to be applied to a wide range of architectures.

## 3  TRAINING WITH DERIVATIVE-CONSTRAINTS

We review an example of training with derivative constraints in the setting of quantum chemistry (Sec. 3.1). Then, we motivate our proposed methods with an experiment demonstrating the difficulty of incorporating gradient constraint information with traditional approaches (Sec. 3.2).

### 3.1  EXAMPLE: POTENTIAL ENERGY SURFACE MODELING

We use potential energy surface (PES) modeling from quantum chemistry as a concrete example to introduce DC training. A PES $U : \mathbb{R}^{3A} \to \mathbb{R}$ gives the energy of a system with $A$ atoms as a function of its atomic coordinates.[1] A PES $U(\mathbf{x})$ and its force field $\mathbf{F}(\mathbf{x})$ can be evaluated by quantum mechanical simulation software such as Gaussian (Frisch et al., 2016) given the 3D Cartesian coordinates $\mathbf{x}$ of the $A$ atoms, *i.e.*, its structure. The force field is the negative gradient of the PES and can be used to simulate the dynamics of the $A$ atoms. In particular, when $\mathbf{F}(\mathbf{x}) = 0$, there are no forces acting on $\mathbf{x}$ which means that the configuration of $\mathbf{x}$ is stable.

Conservation of energy is expressed with the following derivative constraint

$$-\nabla_{\mathbf{x}} U(\mathbf{x}) = \mathbf{F}(\mathbf{x}) \tag{1}$$

that relates the gradient of the PES with the negative force. This connects simulation of a system with its changes in energy. As a result, this means that we can find stable configurations of atomistic

---

[1]We are ignoring symmetries. Technically, $U : \mathbb{R}^{3A-5} \to \mathbb{R}$ for general atomistic systems and $U : \mathbb{R}^{3A-6} \to \mathbb{R}$ when it is planar.

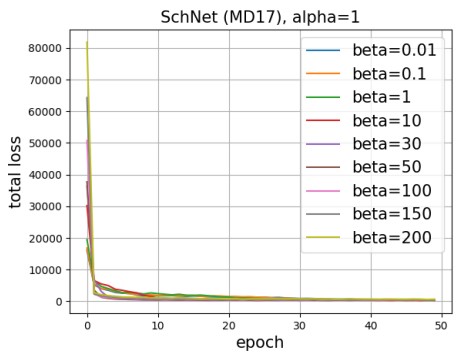 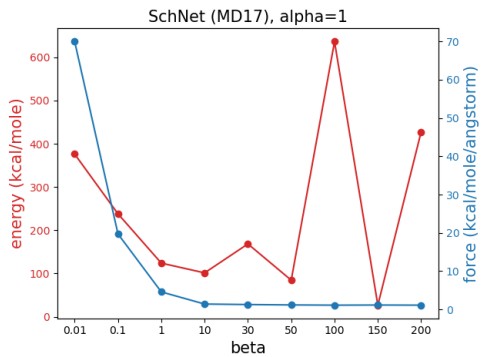

(a) Training loss for $\alpha = 1$ and varying $\beta$.

(b) Energy loss (kcal/mol) and force loss (kcal/(molÅ)) for different $\beta$.

Figure 1: Comparing relative difficulty of learning energy (prediction) versus force (derivative constraint) with SchNet (Schütt et al., 2017) on Asprin molecule in MD17 dataset by varying $\beta$ in loss function (Eq. 2).

systems in nature by finding local minima on the system's PES, since $\mathbf{F}(\mathbf{x}) = -\nabla_{\mathbf{x}} U(\mathbf{x}) = 0$. Consequently, we can study molecules and materials *in silico* if we can model a PES efficiently and accurately enough.

We can construct a surrogate model of a system's PES by fitting a NN $f_\theta$ with parameters $\theta$ to a dataset $\mathcal{D} = \{(\mathbf{x}^i, E^i, F^i)_i : 1 \leq i \leq N\}$ where $\mathbf{x}^i$ are atomic coordinates, $E^i$ is an energy, and $F^i$ are forces – the negative gradient of the energy w.r.t. $\mathbf{x}^i$. This dataset can be created from quantum mechanical simulation software. A surrogate model can be used to accelerate the computation of a PES since quantum mechanical simulation software can be compute-intensive to run. To train a NN on this dataset, we can use the following multi-objective loss function

$$\text{Loss}(f_\theta, \mathcal{D}) = \sum_{i=1}^{N} \alpha \| f_\theta(\mathbf{x}^i) - E^i \|^2 + \beta \| -\nabla_{\mathbf{x}} f_\theta(\mathbf{x}^i) - F^i \|^2 . \tag{2}$$

The terms $\alpha$ and $\beta$ are hyper-parameters that weigh the contributions of the first term involving $f_\theta$'s predictions and the second term involving $f_\theta$'s gradients. Training $f_\theta$ with a gradient-based method will thus involve second-order derivatives. The second term enforces the conservation of energy since it constrains the force prediction of the model to be the observed force. Thus, conservation of energy is violated when training signal from derivative constraints is not efficiently incorporated.

More generally, we can have arbitrary derivatives, constraints, and datasets involving additional supervised signal that contains these constraints for different situations (*e.g.*, thermodynamics, fluid flow). These constraints can come from the underlying partial differential equation (PDE) that describes the natural process. As before, we can add these constraints to a loss term and optimize as before. We refer the reader to the SciML literature (Raissi et al., 2019; Karniadakis et al., 2021; Meng et al., 2022) for more examples.

### 3.2 MOTIVATION

We hypothesize that one fundamental challenge with training a DC model $f_\theta$ is that it is more difficult to incorporate derivative constraint information in the loss term compared to model prediction information in the loss term. As a quick test of our hypothesis and motivation for our methods, we report the following experiment in the setting of quantum chemistry.

**Experiment** We train a selected NN designed for quantum chemistry on a training set consisting of $\{(\mathbf{x}^i, E^i, F^i)_i : 1 \leq i \leq N\}$ tuples for varying $\beta$ values while holding $\alpha = 1$ constant to compare the relative difficulty of predicting the energy versus predicting the force (*i.e.*, involving the gradient). As a reminder, the terms $\alpha$ and $\beta$ control the relative importance of each term in the loss function in Eq. 2. Thus, whether $\alpha > \beta$ or $\alpha < \beta$ can be used as a proxy to determine which

term in the loss is more difficult to learn. In particular, we consider learning of the derivative signal to be more difficult if $\beta > \alpha$ gives lower force loss on the test set compared to energy loss on the test set on a relative basis. We use the relative basis since energies and forces are given in related units, $\frac{\text{kcal}}{\text{mol}}$ versus $\frac{\text{kcal}}{\text{mol}\,\text{Å}}$ respectively, and so direct comparison is not possible.

**Details** In this experiment, we choose a classic NN, Schnet (Schütt et al., 2017). The implementation and hyper-parameters of SchNet are taken from the Open Catalyst Project (OCP) (Chanussot* et al., 2021), a joint effort to between computer scientists and chemistry/material scientists to solve the PES modeling problem. We select the MD17 (Chmiela et al., 2017) dataset which contains $(\mathbf{x}, E^i, F^i)_i$ tuples for 8 different small molecules. As terminology, each $\mathbf{x}^i$ is also called a *conformation*. We train Schnet on $50,000$ conformations of the Asprin molecule for 50 epochs.[2] Instead of normalizing the data before training, we train the networks directly on $\mathbf{x}^i$ so that a surrogate PES can directly predict energies and forces with the same units. The mean energy of Asprin in our training set $-406,737.28\frac{\text{kcal}}{\text{mol}}$ and the variance is $35.36\frac{\text{kcal}^2}{\text{mol}^2}$. The mean force is $423.87\frac{\text{kcal}}{\text{mol}\,\text{Å}}$ and the variance $779.99\frac{\text{kcal}^2}{\text{mol}^2\,\text{Å}^2}$. Thus, the absolute value of the mean energy is roughly 3 orders of magnitude larger than the mean force. We will comment more on this in Sec. 4.2. We use $\beta = \{0.01, 0.1, 1, 10, 30, 50, 100, 200\}$.

**Results** We report training loss curves in Fig. 2a and test loss for various $\beta$ values in Fig. 2b. We observe that the energy loss divided by 1000 (since the energies are roughly 3 orders of magnitude larger) is typically much lower than the force loss. This gives evidence that it is more difficult to incorporate derivative constraint information in the loss term compared to model prediction information in the loss term. Moreover, it is not easy to improve the relative difference, even for large values of $\beta$ which may make the energy loss worse. Finally, we observe that the training losses for each choice of $\beta$ converges to roughly the same level so that there is a trade-off between learning energies and learning forces.

## 4 METHODS

In this section, we propose two ideas to be used in conjunction to improve learning of derivative constraints. First, we propose a new activation function called an integrated ReLU (IReLU) activation function (Sec. 4.1). Second, we introduce *denormalization* and *label rescaling* to help stabilize training of DC NNs with IReLU activations (Sec. 4.2).

### 4.1 INTEGRATED RELU ACTIVATION

The simple observation that we make concerning training a DC NN is that it involves higher-order derivatives. Consequently, higher-order derivatives of activation functions will also be used during the training process. This motivates us to revisit the choice of activation function for DC training since ordinary activation functions have been designed in the setting where only the first-order derivative is used.

We use a simple idea to construct an activation for DC NN training: use an integrated form of an ordinary activation function such as a ReLU when we are performing DC training of NNs. The intuition for doing so is the following: if we only fit derivative constraints, then we should use a ordinary activation (*e.g.*, ReLU) after we have taken the derivative of the original activation in the model. We will discuss this intuition more in Appx. A. Define the *integrated ReLU* (IReLU) to be the activation

$$\text{IReLU}(x) = \int_0^x \text{ReLU}(y)dy = \max(0, 0.5 * x^2). \tag{3}$$

We focus on the IReLU activation in this work since the ReLU is a popular activation. Naturally, the idea of an IReLU can be applied to other activations (Maas et al., 2013; Clevert et al., 2015; Ramachandran et al., 2017; Hendrycks & Gimpel, 2016) as well.

---

[2]We have also trained Schnet for 300 epochs, but have observed fast convergence.

## 4.2 De-Normalization and Label Rescaling

Conventional wisdom is that normalization techniques such as batch normalization may help accelerate the training of NNs as well as improve the stability of training. Notably, centering and rescaling the internal values in a NN might be crucial when using IReLU's since these activations produce higher responses compared to traditional activations. Dataset normalization is also common practice to ensure that input features are set on equal footing. Intuitively, normalization techniques remove the *units* of the features and dataset.

We hypothesize that DC NNs are more sensitive to *units* compared to typical training without derivative constraints because of the linearity of derivatives, *i.e.*, $\nabla f(c\mathbf{x}) = c\nabla f(\mathbf{x})$. We can interpret the constant $c$ as determining the *units* of $\mathbf{x}$, which also determines the units of the derivative $c\nabla f(\mathbf{x})$. In particular, this constant $c$ will appear in the loss term in while training a DC NN and so the loss function is sensitive to the choice of units on the inputs $\mathbf{x}$. We emphasize that a typical setting does not have derivatives of the NN w.r.t. its inputs $\mathbf{x}$, and so these units will not appear in the loss. If our hypothesis holds, then we will need to develop alternative approaches to stabilizing training than the typical unit-insensitive approaches.

Towards this end, we propose two techniques. First, we propose *denormalization*, *i.e.*, the removal of all normalization techniques in a NN architecture. Second, we propose a simple *label rescaling* procedure where we scale the labels in a dataset $\mathcal{D} = (\mathbf{x}^i, \ell_1^i, \ldots \ell_n^i)_i$ by a suitable constant $C$ defined as

$$C = \max_{\ell_j^i}\{C : 0 \leq \frac{\ell_j^i}{C} \leq 1, C \text{ is power of 10 }\}. \tag{4}$$

In the PES modeling example, this means we use the same constant $C$ for both energy and force labels. Intuitively, what label rescaling does is set the units of the model's predictions and derivatives. The loss function in the PES modeling example becomes

$$\sum_i \alpha \|f_\theta(\mathbf{x}^i) - \frac{E^i}{C}\|^2 + \beta\|-\nabla_\mathbf{x} f_\theta(\mathbf{x}^i) - \frac{F^i}{C}\|^2 \tag{5}$$

with label rescaling. Thus, label rescaling plays a similar role to $\alpha$ and $\beta$ in setting units, the difference being that the units are set on the model as opposed to the loss. We emphasize that in label rescaling, we do not normalize the dataset inputs.

## 5 Experiments

We benchmark the performance of our proposed methods on a variety of architectures, datasets, and tasks including quantum chemistry NNs (Sec. 5.1) and PINNs (Sec. 5.2) used in SciML.

### 5.1 Quantum Chemistry

We separate our experiments in quantum chemistry by dataset since different atomistic systems can have different properties. We use the MD17 dataset (Sec. 5.1.1), which has small organic molecules, and OC22 (Chanussot* et al., 2021) (Sec. 5.1.2), which contains large atomistic systems and metals.

### 5.1.1 Experiments on MD17

Our first experiment tests the efficacy of our methods across different architectures for task of potential energy surface modeling. We select SchNet (Schütt et al., 2017), CGCNN Xie & Grossman (2018), ForceNet Hu et al. (2021), DimeNet++Gasteiger et al. (2020b), and GemNet Gasteiger et al. (2021). SchNet and CGCNN are based on a convolutional NN architectures. ForceNet, DimeNet++, and GemNet are based on graph NNs.

We select the MD17 dataset. For each molecule in MD17, we randomly select 50000, 6250, and 6250 conformations from the dataset as training set, validation set and testing set. The molecules include Asprine (Asp.), Benzene (Ben.), Ethanol (Eth.), Malonaldehyde (Mal.), Naphthalene (Nap.), Salicylic acid (Sal.), Toulene (Tol.), and Uracil (Ura.). We present more details in Appx. B.

| Model | Asp. | Ben. | Eth. | Mal. | Nap. | Sal. | Tol. | Ura. |
|---|---|---|---|---|---|---|---|---|
| SchNet | 53.00 | 125.43 | **5.97** | 37.54 | 197.06 | 65.46 | 129.80 | 54.88 |
|  | **1.21** | 0.39 | **0.68** | **1.00** | **0.72** | **1.02** | **0.67** | **0.86** |
| SchNet* | **28.31** | **0.56** | 13.39 | **11.27** | **5.53** | **27.57** | **0.34** | **20.36** |
|  | 1.67 | **0.36** | 1.53 | 1.67 | 1.01 | 1.12 | 1.04 | 1.23 |
| CGCNN | 239.07 | 98.86 | 38.36 | 104.45 | 130.98 | 197.09 | 124.29 | 133.51 |
|  | 14.20 | 6.11 | 8.25 | 14.27 | 8.46 | 8.50 | 9.73 | 9.08 |
| CGCNN* | **137.12** | **16.63** | **4.63** | **30.56** | **66.68** | **78.29** | **36.73** | **81.02** |
|  | **7.13** | **0.61** | **2.96** | **4.99** | **2.38** | **4.77** | **2.91** | **4.93** |
| DimeNet++ | 47.43 | 133.70 | 236.94 | 856.01 | 1096.34 | 580.41 | 301.06 | 669.06 |
|  | 13.82 | 12.45 | 6.41 | 8.87 | 7.82 | 10.41 | 8.64 | 6.15 |
| DimeNet++* | **2.59** | **1.68** | **0.58** | **20.30** | **5.18** | **3.65** | **0.77** | **5.96** |
|  | **2.52** | **0.25** | **0.29** | **0.78** | **0.70** | **1.79** | **0.46** | **0.71** |
| ForceNet | 755.49 | 239.53 | 1048.44 | 874.22 | 1677.71 | 165.17 | 369.33 | 1964.51 |
|  | 21.95 | 117.79 | 17.61 | 31.52 | 18.16 | 22.36 | 18.33 | 46.47 |
| ForceNet* | **13.80** | **19.09** | **3.18** | **4.52** | **5.98** | **41.43** | **5.63** | **14.54** |
|  | **0.89** | **0.33** | **1.15** | **1.36** | **0.35** | **0.34** | **0.50** | **0.26** |
| GemNet | 2201.94 | 352.19 | 470.32 | **5.23** | 564.06 | 1238.93 | 193.50 | 228.10 |
|  | **0.34** | **0.22** | 0.28 | **0.27** | **0.17** | **0.30** | **0.12** | **0.20** |
| GemNet* | **13.16** | **6.31** | **7.10** | 15.36 | **5.87** | **5.67** | **8.13** | **21.98** |
|  | 0.93 | 1.27 | **0.27** | 0.70 | 0.47 | 0.73 | 0.47 | 7.07 |

Table 1: Comparison of model performance trained with original settings and our proposed methods (*). Mean energy loss (kcal/mol) on the upper row and mean force loss (kcal/mol/Å) on the bottom row of the eight molecules in MD17 trained on state-of-the-art models.

Baseline models are trained with the same training configuration and model hyperparameters given by OCP (Chanussot* et al., 2021). We note that Schnet was originally benchmarked on MD17 whereas the other NNs have been tested on other datasets. We train for 50 epochs on the MD17 dataset. Given the fast convergence of the training loss in MD17 (Fig. 2a), we consider 50 epochs is sufficient for models to fully learn the energy and forces. We use a batch size of 20 as recommended in the literature (Chanussot* et al., 2021). We use the Adam optimizer with a learning rate of $0.0001$.

Tab. 1 compares the performance between models trained with original settings and models trained with our proposed methods. We denormalize all networks with normalization layers. For architectures which consist of multiple interaction-output blocks (*e.g.*, DimeNet++ and GemNet), we were only able to replace activation layers in output blocks with IReLU as training with all activations replaced proved to be unstable. For label rescaling, we use the constant $C = 1000000$ since the energies in MD17 are on the order of $400000$. The results of our proposed methods are noted with * in the table. To investigate the individual contribution of IReLU, denormalization, and label rescaling, we also conduct ablation studies (Appx. C). We also experiment on different dataset sizes (Appx. D).

For each architecture, we report the energy loss (upper row) and the force loss (bottom row) separately. We use the units of the original dataset, $\frac{kcal}{mol}$ for energy and $\frac{kcal}{mol \cdot Å}$ for the force respectively. In general, our methods improve upon force loss across most molecules and most architectures. In particular, there is significant improvement in learning forces for CGCNN, DimeNet++ and ForceNet. We observe cases where our method performs worse on forces (*e.g.*, SchNet and GemNet) but provides competitive performance. Perhaps surprisingly, our methods also improve the energy loss (38 out of 40 cases). We might reason that better incorporating force information would lead to improved learning of the physics. Nevertheless, it would be an interesting direction of future work to study this in more detail.

### 5.1.2 EXPERIMENTS ON OC22 DATASET

To validate our methods on more datasets, we also compare the performance with and without our proposed methods on OC22 (Chanussot* et al., 2021). OC22 contains 62331 relaxations of oxides calculated at the DFT level. It contains a wide range of crystal structures (*e.g.*, monoclinic,

| Model | Energy MAE Loss (eV) | Force MAE Loss (eV/Å) |
|---|---|---|
| SchNet | **6.94** | **0.10** |
| SchNet* | 22.4621 | 0.12 |
| CGCNN | 233.90 | 0.32 |
| CGCNN* | **71.33** | **0.08** |
| DimeNet++ | 5.10 | 0.09 |
| DimeNet++* | **0.57** | **0.01** |
| ForceNet | **4.48** | 0.33 |
| ForceNet* | 5.90 | **0.10** |

Table 2: Comparison of performance on OC22 with original settings and our proposed methods.

tetragonal) that contain heavier elements (*e.g.*, metalloids, transition metals). Compared to MD17, OC22 consists of various large structure/metals (more than 100 atoms) mixed together in the training, validation, and testing set. In this dataset, the absolute value of the mean energy is only 1 order of magnitude larger than the mean force (compared to 3 in MD17). We randomly select 200000 molecules in OC22 training split as our training set and 25000 molecules in OC22 validation (out of domain) split as our testing set.

Baseline models are trained with the same training configuration and model hyperparameters given by OCP (Chanussot* et al., 2021). We train all models for 50 epochs. We use a batch size of 20 as recommended in the literature (Chanussot* et al., 2021) for SchNet and CGCNN. Due to hardware memory limitations we tested DimeNet++ and ForceNet with a batch size 10. We were not able to test GemNet due to hardware limitations. We use the Adam optimizer with a learning rate of 0.00001 for ForceNet and 0.0001 for others.

Tab. 2 shows that our methods produce better force predictions for CGCNN, DimeNet++, and ForceNet. In those models where we improved force losses, CGCNN and DimeNet++ also improve energy losses. It would be interesting to investigate why SchNet with our methods performs worse on OC22. As a reminder, both SchNet and CGCNN are based on convolutional architectures, and CGCNN's performance is improved with our method. We emphasize again that we do not modify the given architectures beyond replacing the activation functions (when appropriate).

## 5.2 Physics-informed Neural Networks

To validate the generalization ability of our proposed methods in other domains aside from quantum chemistry, we also experiment on physics-informed neural networks (PINNs) (Raissi et al., 2019; Karniadakis et al., 2021; Wu et al., 2018). PINNs are a general family of models that use a NN to predict the solution of a partial differential equation (PDE). The solution of a PDE is a latent function $\psi(x, t)$ that describe physical measurements (*e.g.*, temperature and velocity) as a function of spatial coordinates $x$ and time $t$. PINNs enforce physical constraints on the solution $\psi(x, t)$ by enforcing that the solution satisfies the governing PDE in the loss function.

In general the loss function for a PINNs takes the form below

$$\mathcal{L}(\psi, \mathcal{D}) = \mathcal{L}_f(\psi, \mathcal{D}_f) + \mathcal{L}_{ICBC}(\psi, \mathcal{D}_{IC}, \mathcal{D}_{BC}) \tag{6}$$

where $\psi$ is a learned PDE solution (*e.g.*, a NN) and $\mathcal{D} = (\mathcal{D}_f, \mathcal{D}_{IC}, \mathcal{D}_{BC})$ is a dataset consisting of several components containing additional constraints. The first term

$$\mathcal{L}_f(\psi, \mathcal{D}_f) = \frac{1}{|\mathcal{D}_f|} \sum_{(\mathbf{x}, t, \mathbf{y}) \in \mathcal{D}_f} \mathcal{F}\left(\frac{\partial \psi(\mathbf{x}, t)}{\partial \mathbf{x}}, \frac{\partial \psi(\mathbf{x}, t)}{\partial t}, \frac{\partial^2 \psi(\mathbf{x}, t)}{\partial \mathbf{x}^2}, \frac{\partial^2 \psi(\mathbf{x}, t)}{\partial \mathbf{x} \partial t}, \dots, \mathbf{y}\right) \tag{7}$$

gives the predicted solution's loss evaluated on a spatial-temporal grid $(\mathbf{x}, t) \in \mathcal{D}_f$ using a function $\mathcal{F}$. The loss is a function of additional derivatives of the predicted solution $\psi$ w.r.t. its inputs according to the governing PDE. Thus, the loss function for a PINN may contain many higher-order derivatives.

The second term

$$\mathcal{L}_{ICBC}(\psi, \mathcal{D}_{IC}, \mathcal{D}_{BC}) = \frac{1}{|\mathcal{D}_{IC}|} \sum_{(\boldsymbol{x}, \mathbf{i}) \in \mathcal{D}_{IC}} \mathcal{I}(\psi(\mathbf{x}, 0), \mathbf{i}) + \frac{1}{|\mathcal{D}_{BC}|} \sum_{(\boldsymbol{x}, t, \mathbf{b}) \in \mathcal{D}_{BC}} \mathcal{B}(\psi(\mathbf{x}, t), \mathbf{b}) \tag{8}$$

| Method | MSE | $\mathcal{L}_f^\dagger$ | $\mathcal{L}_{IC}$ | $\mathcal{L}_{BC}$ |
|---|---|---|---|---|
| Tanh + BN | 1.70 | 8e-16 | 5e-5 | 3e-5 |
| IReLU + BN | 2.42 | 9e-12 | 8e-6 | 2e-6 |
| Tanh (original) | 1.59 | 1e-5 | **3e-6** | 6e-6 |
| IReLU (ours) | **0.99** | **<e-45** | 0.08 | **<e-45** |

Table 3: MSE and loss terms of the Advection equation. $\mathcal{L}_f$ is the loss of the PDE which governs the Advection equation, $\mathcal{L}_{IC}$ is the loss on the initial conditions, and $\mathcal{L}_{BC}$ is the loss on the boundary conditions.

enforces constraints on the PDE solution given by initial conditions ($\mathcal{D}_{IC}$) and boundary conditions ($\mathcal{D}_{BC}$). $\mathcal{I}$ and $\mathcal{B}$ are the respective loss functions for the initial conditions and boundary conditions. These conditions further constrain the solution of a PDE. There can be multiple IC and BC loss terms, Moreover, the IC and BC loss terms (not shown) can also involve higher-order derivatives in certain PINNs.

For our experiments with PINNs, we adapt baseline architectures, datasets and training configurations from PDEBench (Takamoto et al., 2022a;b). PDEBench provides implementations and benchmarks of SciML models including PINNs for learning (1) the Advection equation (Sec. 5.2.1), (2) the compressible fluid dynamic equation (Sec. 5.2.2), and (3) the diffusion-reaction equation (Sec. 5.2.3). In the baseline architecture, all PINNs use the same MLP (multi-layer perceptron) architecture with 6 hidden layers of 40 neurons in each to simulate their latent function $\psi$. The MLP uses Tanh activation function for every hidden layer. We use the same library DeepXDE (Lu et al., 2021) to construct and train the backbone MLP.

For all PDES, we compare training a PINN to learn the PDE with activations replaced with IReLU activations. We did not find a need for label rescaling. For comparison, we also add in batch normalization (BN) to study its impact. Following PINN convention, we measure the model performance by using the mean square error (MSE) of the PINN's latent function prediction (*i.e.*, $\psi(x,t)$). Thus, the loss terms which involve higher-order derivatives are taken purely as constraints. Loss terms labeled with $\dagger$ are the terms which involve derivatives of the model w.r.t its inputs. To give more fine-grained information about how our methods impact each component of the loss term, we provide the MSE value of each loss term evaluated in the last epoch of training along side the predictive MSE evaluated in testing set.

### 5.2.1 ADVECTION EQUATION

The Advection equation has a simple PDE that involves first-order partial derivatives of the model w.r.t. its input in $\mathcal{L}_f$, one initial condition in $\mathcal{L}_{ICBC}$, and one boundary condition in $\mathcal{L}_{ICBC}$. Thus, this task tests our method's performance on loss functions in DC training with 3 terms. The full loss function associated with the Advection equation is presented in the Appx. E.1. Both standard training and training with our methods use 15000 epochs on the 1D Advection dataset with the Adam optimizer and an initial learning rate of 0.001 following PDEBench.

Tab. 3 presents the predictive loss evaluated on the test set and the training loss of each term in the loss function evaluated in the last epoch of training since these terms act as constraints. The model trained with our method achieves the best predictive loss ($\mathcal{L}_f$) on the test set. We also improve two of the training loss terms. It is also interesting to observe that adding batch normalization (BN) decreases performance.

### 5.2.2 COMPRESSIBLE FLUID DYNAMICS EQUATION

The compressible fluid dynamics (CFD) equation contains 6 total initial and boundary conditions in $\mathcal{L}_{ICBC}$. Like the Advection equation, it also involves first-order derivatives of the model w.r.t. its input in $\mathcal{L}_f$. Thus, this task tests our method's ability to handle many terms in the loss function. The full loss function associated with the CFD equation is presented in the Appx. E.2. Both standard training and training with our methods use 15000 epochs on the 1D CFD dataset from PDEBench with the Adam optimizer and an initial learning rate of 0.001 following PDE bench.

| Method | MSE | $\mathcal{L}_f^\dagger$ | $\mathcal{L}_{IC_p}$ | $\mathcal{L}_{IC_d}$ | $\mathcal{L}_{IC_v}$ | $\mathcal{L}_{BC_p}$ | $\mathcal{L}_{BC_d}$ | $\mathcal{L}_{BC_v}$ |
|---|---|---|---|---|---|---|---|---|
| Tanh + BN | 1.11 | 271.94 | 2910.32 | 11.37 | 0.25 | 0.15 | 0.15 | 0.15 |
| IReLU + BN | 1.66 | 2e+7 | 4656.73 | 32.97 | 0.87 | 1.96 | 1.96 | 1.96 |
| Tanh (original) | 0.87 | 7.96 | 4694.60 | 16.01 | 0.27 | 0.11 | 0.11 | 0.11 |
| IReLU (ours) | **0.40** | **0.11** | **88.78** | **0.21** | **0.17** | **1e-6** | **1e-6** | **1e-6** |

Table 4: MSE and loss terms of the CFD equation. $\mathcal{L}_f$ is the loss of the PDE which governs the CFD equation. $\mathcal{L}_{IC_p}$, $\mathcal{L}_{IC_d}$, and $IC_v$ are the loss on the initial conditions corresponding to pressure, density, and velocity respectively. $\mathcal{L}_{BC_p}$, $\mathcal{L}_{BC_d}$, and $\mathcal{L}_{BC_v}$ are the loss on the boundary conditions corresponding to pressure, density, and velocity respectively.

| Method | MSE | $\mathcal{L}_f^\dagger$ | $\mathcal{L}_{IC_u}$ | $\mathcal{L}_{IC_v}$ | $\mathcal{L}_{BC}^\dagger$ | $\mathcal{L}_{BC_u}$ | $\mathcal{L}_{BC_v}$ |
|---|---|---|---|---|---|---|---|
| Tanh + BN | 4.94 | 0.46 | 1.18 | 1.06 | **2e-15** | 0.08 | 0.008 |
| IReLU + BN | 8.03 | 7e+4 | 0.98 | 1.00 | 2e-10 | 4.44 | 4.34 |
| Tanh (original) | 1.35 | 2e-4 | **0.97** | **0.99** | 1e-4 | **1e-4** | **3e-5** |
| IReLu (ours) | **1.13** | **1e-4** | 0.98 | 1.00 | 3e-14 | 0.004 | 3e-4 |

Table 5: MSE and loss terms of DR equation. $\mathcal{L}_f$ is the loss of the PDE which governs the DR equation. $\mathcal{L}_{IC_u}$ and $\mathcal{L}_{IC_v}$ are the loss on the initial conditions corresponding to the activator and the inhibitor respectively. $\mathcal{L}_{BC}$, $\mathcal{L}_{BC_u}$, and $\mathcal{L}_{BC_v}$ give the loss on the boundary conditions corresponding to the derivative of the boundary conditions, the activator, and the inhibitor respectively.

Tab. 4 presents the predictive loss $\mathcal{L}_f$ evaluated on the test set. We also include the training loss of each term in $\mathcal{L}_{ICBC}$ evaluated in the last epoch of training. The model trained with IReLU and denormalization performs the best on all training terms. Our methods perform exceptionally well on the term involving derivatives ($\mathcal{L}_f$ loss). As before, we also observe that adding BN decreases performance.

### 5.2.3 DIFFUSION REACTION EQUATION

The diffusion-reaction (DR) equation involves the most complex PDE in terms of derivative constraints. The loss on the PDE solution's prediction contains second-order derivatives, which means that third-order derivative information is used during NN training. Additionally, there are boundary conditions that utilize gradient information to describe the solution's boundary state at each given time step. Thus, this task tests our method's ability to cope with higher-order derivatives and derivative constraints in multiple terms. The full loss function associated with the diffusion reaction equation is presented in the Appx. E.3. For the DR equation, both standard training and training with our methods use 100 epochs on the 2D DR dataset (1000 samples) from PDEBench with the Adam optimizer and an initial learning rate of 0.001.

Tab. 5 presents the MSE loss on the testing set and training loss of each term in $\mathcal{L}_{ICBC}$ in the last epoch of training. The models with IReLU and denormalization have the best predictive loss, which involves second-order derivative information. Additionally, we also achieve competitive performance on the boundary conditions that involve derivatives. This experiment demonstrates that our proposed methods has ability to fit gradients of NN that have order more than two. Note that the IReLU has a vanishing third-order derivative so other integrated activations may perform better.

## 6 CONCLUSION

In this paper, we introduce IReLU activations, denormalization, and label rescaling to improve DC training of NNs. We demonstrate that these methods improve the performance across a range of tasks and architectures in the setting of training with derivative constraints. It would be interesting to further investigate the applicability of these methods to more domains. It would also be an interesting direction of future work to investigate novel regularization techniques that work in this setting.

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

# A  WHY INTEGRATED ReLU?

We give intuition for the IReLU activation by walking through the difference between training without derivative-constraints and with derivative constraints. In the following, let $\sigma$ be an activation function and $f(\mathbf{x}, \theta)$ be a NN layer that takes an input $\mathbf{x}$ with parameters $\theta$.

**Update without derivative constraints**  During backpropagation, the parameters $\theta$ will be updated using the partial derivative w.r.t. $\theta$ as

$$\frac{\partial}{\partial \theta}(\sigma(f(\mathbf{x}, \theta))) = \sigma'(f(\mathbf{x}, \theta))\frac{\partial f(\mathbf{x}, \theta)}{\partial \theta} \ . \tag{9}$$

**Update with derivative constraints**  When we are training with derivative constraints encoded in the loss function, we additionally need to take derivatives of the model w.r.t. its inputs. Consequently, during backpropagation, the parameters $\theta$ will be updated using an additional term

$$\frac{\partial^2}{\partial \mathbf{x} \partial \theta}(\sigma(f(\mathbf{x}, \theta))) = \sigma''(f(\mathbf{x}, \theta))\frac{\partial f(\mathbf{x}, \theta)}{\partial \mathbf{x}}\frac{\partial f(\mathbf{x}, \theta)}{\partial \theta} + \sigma'(f(\mathbf{x}, \theta))\frac{\partial^2 f(\mathbf{x}, \theta)}{\partial \mathbf{x} \partial \theta} \tag{10}$$

when the first-order derivative of the model w.r.t. its inputs is used. For an activation $\sigma = \mathrm{ReLU}$, we note that $\mathrm{ReLU}''(x) = 0$ so that the first term vanishes, leaving us with

$$\frac{\partial^2}{\partial \mathbf{x} \partial \theta}(\mathrm{ReLU}(f(\mathbf{x}, \theta))) = \mathrm{ReLU}'(f(\mathbf{x}, \theta))\frac{\partial^2 f(\mathbf{x}, \theta)}{\partial \mathbf{x} \partial \theta} \ . \tag{11}$$

Consequently, a standard activation function such as ReLU loses training signal in the DC setting.

In a multi-objective loss function where prediction errors and derivative constraints also introduce error, the total update has contributions from both updates. For example, the loss function in Equation 2 in the setting of quantum chemistry contains both energy and force terms, so the total update is given as

$$\frac{\partial}{\partial \theta}(\sigma(f(\mathbf{x}, \theta))) + \frac{\partial^2}{\partial \mathbf{x} \partial \theta}(\sigma(f(\mathbf{x}, \theta))) =$$
$$\left(\sigma'(f(\mathbf{x}, \theta)) + \sigma''(f(\mathbf{x}, \theta))\frac{\partial f(\mathbf{x}, \theta)}{\partial \mathbf{x}}\right)\frac{\partial f(\mathbf{x}, \theta)}{\partial \theta} + \sigma'(f(\mathbf{x}, \theta))\frac{\partial^2 f(\mathbf{x}, \theta)}{\partial \mathbf{x} \partial \theta} \tag{12}$$

where we have regrouped the terms for comparison with the original activation. Under the assumption that $\frac{\partial^2 f(\mathbf{x}, \theta)}{\partial \mathbf{x} \partial \theta} \approx \frac{\partial f(\mathbf{x}, \theta)}{\partial \mathbf{x}}\frac{\partial f(\mathbf{x}, \theta)}{\partial \theta}$, we see that the total update is

$$\left(\sigma'(f(\mathbf{x}, \theta)) + (\sigma'(f(\mathbf{x}, \theta)) + \sigma''(f(\mathbf{x}, \theta)))\frac{\partial f(\mathbf{x}, \theta)}{\partial \mathbf{x}}\right)\frac{\partial f(\mathbf{x}, \theta)}{\partial \theta} \ . \tag{13}$$

The relative ratio of the updates given by standard training versus DC training is thus

$$1/\left(2 + \frac{\sigma''(f(\mathbf{x}, \theta))}{\sigma'(f(\mathbf{x}, \theta))}\frac{\partial f(\mathbf{x}, \theta)}{\partial \mathbf{x}}\right) \ . \tag{14}$$

We note that the importance of the contribution of the derivative $\frac{\partial f(\mathbf{x}, \theta)}{\mathbf{x}}$ is given by the ratio of $\sigma''$ compared to $\sigma'$. Thus, we can specifically control the contribution of derivative constraints by adjusting the activation.

# B  MD17 MOLECULE DETAILS

In Tab. 6, we provide the details of the 8 molecules in MD17. MD17 is a collection of conformations of small molecules consisting of Carbon (C), Hydrogen (H), Nitrogen (N), and Oxygen (O) atoms.

# C  ABLATION STUDIES

In this section, we show the results of ablation studies on IReLU activations, denormalization, and label rescaling with the same methodology mentioned in Sec. 5.1.

| Abbreviation | Molecule | Formula | Num Atoms | Num Conformations |
|---|---|---|---|---|
| Asp. | Aspirin | $C_9H_8O_4$ | 21 | 211762 |
| Ben. | Benzene | $C_6H_6$ | 12 | 627983 |
| Eth. | Ethanol | $C_2H_6O$ | 9 | 555092 |
| Mal. | Malonaldehyde | $C_3H_4O_2$ | 9 | 993237 |
| Nap. | Naphthalene | $C_{10}H_8$ | 18 | 326250 |
| Sal. | Salicylic acid | $C_7H_6O_3$ | 16 | 320231 |
| Tol. | Toluene | $C_6H_5CH_3$ | 15 | 442790 |
| Ura. | Uracil | $C_4H_4N_2O_2$ | 12 | 133770 |

Table 6: Details of molecules in MD17

| Model | Asp. | Ben. | Eth. | Mal. | Nap. | Sal. | Tol. | Ura. |
|---|---|---|---|---|---|---|---|---|
| SchNet | 53.00 | 125.43 | 5.97 | 37.54 | 197.06 | 65.46 | 129.80 | 54.88 |
|  | **1.21** | 0.39 | **0.68** | **1.00** | **0.72** | **1.02** | **0.67** | **0.86** |
| SchNet* | **12.35** | **2.38** | **0.84** | **1.25** | **1.97** | **8.22** | **1.88** | **2.47** |
|  | 13.39 | **0.34** | 1.43 | 1.69 | 2.12 | 6.14 | 1.72 | 1.85 |
| CGCNN | 239.07 | 98.86 | **38.36** | 104.45 | 130.98 | 197.09 | 124.29 | 133.51 |
|  | **14.20** | **6.11** | **8.25** | **14.27** | **8.46** | **8.50** | **9.70** | **9.08** |
| CGCNN* | **97.41** | **12.75** | 39.28 | **93.49** | **113.61** | **109.16** | **30.70** | **65.45** |
|  | 26.90 | 37.05 | 12.45 | 20.87 | 14.94 | 35.32 | 15.54 | 18.55 |
| DimeNet++ | **47.43** | 133.70 | 236.94 | 856.01 | 1096.34 | 580.41 | 301.06 | 669.06 |
|  | **13.82** | 12.45 | 6.41 | **8.87** | **7.82** | **10.41** | **8.64** | **6.15** |
| DimeNet++* | 298.71 | **48.13** | **39.76** | 90.78 | 109.36 | 127.91 | 74.38 | 119.00 |
|  | 26.75 | **4.50** | **4.69** | 11.52 | 13.40 | 20.14 | 12.56 | 21.87 |
| ForceNet | **755.49** | **239.53** | 1048.44 | **874.22** | 1677.71 | **165.17** | **369.33** | **1964.51** |
|  | **21.95** | 117.79 | **17.61** | **31.52** | **18.16** | **22.36** | **18.33** | **46.47** |
| ForceNet* | 4e+5 | 955.28 | **593.23** | 1085.65 | **940.71** | 783.66 | 1181.15 | 2129.21 |
|  | 32.34 | **10.34** | 74.83 | 654.96 | 24.30 | 1963.86 | 154.05 | 1617.17 |
| GemNet | 2201.94 | **352.19** | 470.32 | **5.23** | **564.06** | **1238.93** | 193.50 | 228.10 |
|  | **0.34** | **0.22** | **0.28** | **0.27** | **0.17** | **0.30** | **0.12** | **0.20** |
| GemNet* | **44.31** | NaN | **13.06** | NaN | NaN | NaN | **10.26** | **115.43** |
|  | 1.81 | NaN | 0.88 | NaN | NaN | NaN | 0.18 | 1.57 |

Table 7: Comparison of model performance trained with original activation and with Integrate ReLU. Mean energy loss (kcal/mol) on the upper row and mean force loss (kcal/mol/Å) on the bottom row of the eight molecules in MD17 trained on state-of-the-art models. The models are trained on single molecule training set in MD17 and tested on their corresponding testing set.

## C.1 ABLATION STUDY: INTEGRATED RELU

In Sec. 5.1.1, we show that our methods generally reduce both energy and force error. To investigate the respective contribution of the IReLU activation, we conduct an ablation experiment on IReLU. In Tab. 7, we compare between the state-of-the-art models performance trained with their original activation function and trained with IReLU (marked with *). SchNet uses Shifted-Softplus, CGCNN uses Softplus, and DimeNet++, ForceNet, and GemNet uses SiLU in their original architectures.

Tab. 7 shows that applying IReLU activations exclusively to the models does not consistently improve energy loss or force loss. Due to the squaring operation (Eq. 3) in an IReLU activation, values in features may diverge, resulting in numeric instability in the training phase. GemNet is an example that suffers from such exploding features. We report NaN when this happens in our experiments.

To show that IReLU activations still benefits model performance (with an appropriate regularization technique) in the results of Sec. 1, we also conduct this ablation experiment when denormalization and label rescaling are applied. We present results in Tab. 8. In this table, we show that IReLU largely helps reducing energy losses and also reduce force losses in most cases.

| Model | Asp. | Ben. | Eth. | Mal. | Nap. | Sal. | Tol. | Ura. |
|---|---|---|---|---|---|---|---|---|
| SchNet | 610.62 | 5e+4 | 205.45 | 77.42 | 8218.45 | 1378.72 | 2e+4 | 76.96 |
| | 9.57 | 0.90 | 6.35 | 4.87 | **0.81** | 9.25 | 3.23 | 4.42 |
| SchNet* | **28.31** | **0.56** | **13.39** | **11.27** | **5.53** | **27.57** | **0.34** | **20.36** |
| | **1.67** | **0.36** | **1.53** | **1.67** | 1.01 | **1.12** | **1.04** | **1.23** |
| CGCNN | 816.04 | 1000.86 | 496.82 | 217.76 | 870.55 | 147.07 | 1903.14 | 456.21 |
| | **1.27** | **0.35** | **0.61** | **0.73** | **0.80** | **1.01** | **0.84** | **1.12** |
| CGCNN* | **137.12** | **16.63** | **4.63** | **30.56** | **66.68** | **78.29** | **36.73** | **81.02** |
| | 7.13 | 0.61 | 2.96 | 4.99 | 2.38 | 4.77 | 2.91 | 4.93 |
| DimeNet++ | 237.34 | 235.38 | 70.11 | 417.45 | 101.76 | 157.36 | 295.44 | 240.70 |
| | **0.75** | 1.34 | **0.24** | **0.32** | **0.53** | **0.58** | 0.76 | **0.42** |
| DimeNet++* | **2.59** | **1.68** | **0.58** | **20.30** | **5.18** | **3.65** | **0.77** | **5.96** |
| | 2.52 | **0.25** | 0.29 | 0.78 | 0.70 | 1.79 | **0.46** | 0.71 |
| ForceNet | 882.29 | 625.52 | 233.99 | 100.58 | 727.00 | 673.01 | 289.86 | 1366.66 |
| | 1.37 | 0.77 | **0.68** | 1.59 | 0.94 | 0.95 | 1.02 | 0.82 |
| ForceNet* | **13.80** | **19.09** | **3.18** | **4.52** | **5.98** | **41.43** | **5.63** | **14.54** |
| | **0.89** | **0.33** | 1.15 | **1.36** | **0.35** | **0.34** | **0.50** | **0.26** |
| GemNet | 220.03 | 718.87 | 153.32 | 329.15 | 681.90 | 1926.48 | 117.86 | 555.19 |
| | 6.57 | **0.71** | 0.38 | 1.90 | 0.50 | 4.22 | 0.64 | **2.69** |
| GemNet* | **13.16** | **6.31** | **7.10** | **15.36** | **5.87** | **5.67** | **8.13** | **21.98** |
| | **0.93** | 1.27 | **0.27** | **0.70** | **0.47** | **0.73** | **0.47** | 7.07 |

Table 8: Comparison of model performance trained with original activation and with Integrate ReLU when denormalization and label rescaling are used. Mean energy loss (kcal/mol) on the upper row and mean force loss (kcal/mol/Å) on the bottom row of the eight molecules in MD17 trained on state-of-the-art models. The models are trained on single molecule training set in MD17 and tested on their corresponding testing set.

| Model | Asp. | Ben. | Eth. | Mal. | Nap. | Sal. | Tol. | Ura. |
|---|---|---|---|---|---|---|---|---|
| CGCNN | **239.07** | **98.86** | **38.36** | **104.45** | **130.98** | 197.09 | **124.29** | **133.51** |
| | 14.20 | 6.11 | 8.25 | 14.27 | 8.46 | 8.50 | 9.70 | 9.08 |
| CGCNN* | 816.04 | 1000.86 | 496.82 | 217.76 | 870.55 | **147.07** | 1903.14 | 456.21 |
| | **1.27** | **0.35** | **0.61** | **0.73** | **0.80** | **1.01** | **0.84** | **1.12** |
| ForceNet | **755.49** | **239.53** | 1048.44 | 874.22 | 1677.71 | **165.17** | 369.33 | 1964.51 |
| | 21.95 | 117.79 | 17.61 | 31.52 | 18.16 | 22.36 | 18.33 | 46.47 |
| ForceNet* | 882.29 | 625.52 | **233.99** | **100.58** | **727.00** | 673.01 | **289.86** | **1366.66** |
| | 1.37 | **0.77** | **0.68** | **1.59** | **0.94** | **0.95** | **1.02** | **0.82** |

Table 9: Comparison of model performance trained with and without denormalization and label rescaling. Mean energy loss (kcal/mol) on the upper row and mean force loss (kcal/mol/Å) on the bottom row of the eight molecules in MD17 trained on CGCNN and ForceNet. The models are trained on single molecule training set in MD17 and tested on their corresponding testing set.

## C.2 ABLATION STUDY: DENORMALIZATION AND LABEL RESCALING

We also evaluate the contribution of denormalization and rescaling. In Tab. 9, we show the comparison of performance between original models and denormalized models. We choose CGCNN and ForceNet to experiment on since they originally contain normalization layers in their respective architectures. For the denormalized models, we also apply label rescaling with constant $C = 1000000$ to stabilize the training phase. We use * to indicate the denormalized models trained with rescaled labels in the table.

From Tab. 9, we can see that the improved models consistently reduce force errors by a large amount. Meanwhile, energy losses show close preference for original models. Overall, denormalization and label rescaling significantly improves force losses at the cost of producing higher energy losses. We can improve this with the IReLU activation as we demonstrated in Tab. 1.

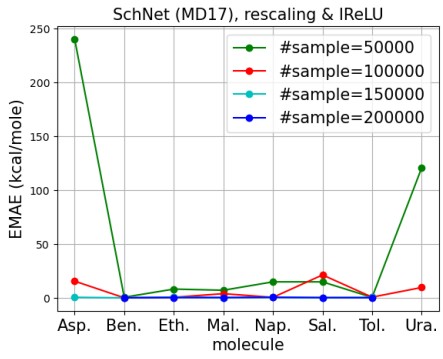 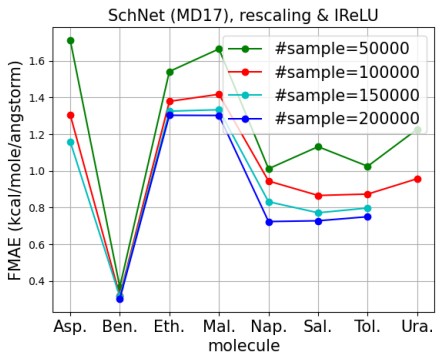

(a) Energy test loss for different training set sizes.

(b) Force test loss for different training set sizes.

Figure 2: Test losses of our methods as a function of dataset size. We do not have enough Asprin or Uracil conformations to test at larger dataset sizes.

## D SCALING ON DATA

One nice property of NNs that has been demonstrated empirically is that their performance improves with increasing data. Because we change a fundamental part of a NN architecture, namely the activation function, we test that this scaling property still holds. We train SchNet with IReLU activations and label rescaling ($C = 1000000$) on $\{50000, 100000, 150000, 200000\}$ datapoints from the MD17 dataset.[3]. Fig. 2 reports that we can reduce both energy loss and force loss as we increase the dataset size.

## E LOSS FUNCTIONS OF PINNS

In this section, we explain the PINNs and their respective loss function used during training in Sec. 5.2.

### E.1 ADVECTION EQUATION

The Advection equation is a PDE that describes the motion of fluid in a given velocity vector field. With a solenoidal velocity vector field, we have:

$$\psi_t + \mathbf{u} \cdot \nabla \psi = 0 \tag{15}$$

where in the 1D case, $\mathbf{u} \cdot \nabla \psi = u_x \psi_x$. In this dataset, initial conditions $\psi(x, 0)$ are set as a superposition of two sinusoidal waves which coordinates are picked randomly and boundary conditions are periodic. Therefore, we use the following loss function derived from these constraints in our experiment:

$$\mathcal{L}\left(\boldsymbol{\psi}; \mathcal{D}\right) = \frac{1}{|\mathcal{D}_f|} \sum_{\mathbf{x} \in \mathcal{D}_f} \|\psi_t(\mathbf{x}) + \beta \psi_x(\mathbf{x})\|_2^2 + \frac{1}{|\mathcal{D}_{ic}|} \sum_{\mathbf{x}, \mathbf{y} \in \mathcal{D}_{ic}} \|\psi(\mathbf{x}) - \mathbf{y}\|_2^2 + \frac{1}{|\mathcal{D}_{bc}|} \sum_{\mathbf{x}, \mathbf{y} \in \mathcal{D}_{bc}} \|\psi(\mathbf{x}) - \mathbf{y}\|_2^2 \tag{16}$$

where training set $\mathcal{D} = \mathcal{D}_f + \mathcal{D}_{ic} + \mathcal{D}_{bc}$. We use $\beta = 0.1$ in our experiments.

### E.2 COMPRESSIBLE FLUID DYNAMIC EQUATION

The compressible fluid dynamic (CFD) equation (*i.e.*, compressible Navier-Stokes equation), are PDEs that express momentum balance an conservation of mass for Newtonian fluids. The CFD

---

[3]Limited by the amount of conformations provided in MD17, the Aspirin trajectory does not have enough sample to test 150000 and 200000 points. The Uracil trajectory does not have enough samples to test on 200000 points.

equation relates pressure, temperature and density.

$$\rho_t + \nabla \cdot (\rho \mathbf{v}) = 0 \tag{17}$$

$$\rho (\mathbf{v}_t + \mathbf{v} \cdot \nabla \mathbf{v}) = -\nabla p + \eta \Delta \mathbf{v} + (\zeta + \eta/3)\nabla(\nabla \cdot \mathbf{v}) \tag{18}$$

$$\left[\epsilon + \frac{\rho v^2}{2}\right]_t + \nabla \cdot \left[\left(\epsilon + p + \frac{\rho v^2}{2}\right)\mathbf{v} - \mathbf{v} \cdot \sigma'\right] = 0 \tag{19}$$

where $\rho$ is the density, v is the velocity, p is the pressure, $\epsilon = 1.5 * p$, $\sigma$ is the viscous stress tensor, $\eta$ is the shear viscosity and $\zeta$ is the bulk viscosity. In this dataset, initial conditions $\psi(x, 0)$ are set as a super-position of four sinusoidal waves which coordinates are picked randomly and boundary conditions are outgoing, which allows waves and fluid to escape from the computational domain. Therefore, we use the following loss function derived from these constraints in our experiment:

$$\mathcal{L}(\boldsymbol{\psi}; \mathcal{D}) = \frac{1}{|\mathcal{D}_f|} \sum_{\mathbf{x} \in \mathcal{D}_f} \|f_1(\mathbf{x}) + f_2(\mathbf{x}) + f_3(\mathbf{x})\|_2^2 +$$

$$\frac{1}{|\mathcal{D}_{ic_d}|} \sum_{\mathbf{x},\mathbf{y} \in \mathcal{D}_{ic_d}} \|d(\mathbf{x}) - \mathbf{y}\|_2^2 + \frac{1}{|\mathcal{D}_{ic_v}|} \sum_{\mathbf{x},\mathbf{y} \in \mathcal{D}_{ic_v}} \|v(\mathbf{x}) - \mathbf{y}\|_2^2 + \frac{1}{|\mathcal{D}_{ic_p}|} \sum_{\mathbf{x},\mathbf{y} \in \mathcal{D}_{ic_p}} \|p(\mathbf{x}) - \mathbf{y}\|_2^2 +$$

$$\frac{1}{|\mathcal{D}_{bc_d}|} \sum_{\mathbf{x},\mathbf{y} \in \mathcal{D}_{bc_d}} \|d(\mathbf{x}) - \mathbf{y}\|_2^2 + \frac{1}{|\mathcal{D}_{bc_v}|} \sum_{\mathbf{x},\mathbf{y} \in \mathcal{D}_{bc_v}} \|v(\mathbf{x}) - \mathbf{y}\|_2^2 + \frac{1}{|\mathcal{D}_{bc_p}|} \sum_{\mathbf{x},\mathbf{y} \in \mathcal{D}_{bc_p}} \|p(\mathbf{x}) - \mathbf{y}\|_2^2 \tag{20}$$

We use $\eta = 10^{-8}$ and $\zeta = 10^{-8}$.

$$f_1(\mathbf{x}) = \rho_t(\mathbf{x}) + (\rho v)_x(\mathbf{x}) \tag{21}$$

$$f_2(\mathbf{x}) = \rho(\mathbf{x}) * (v_t(\mathbf{x}) + v(\mathbf{x}) * v_x(\mathbf{x})) - p_x(\mathbf{x}) \tag{22}$$

$$f_3(\mathbf{x}) = [p/(\gamma - 1) + 0.5 * h * u^2]_t(\mathbf{x}) + [u * (p/(\gamma - 1) + 0.5 * h * u^2) + p)]_x(\mathbf{x}) \tag{23}$$

where training set $\mathcal{D} = \mathcal{D}_f + \mathcal{D}_{ic_d} + \mathcal{D}_{ic_v} + \mathcal{D}_{ic_p} + \mathcal{D}_{bc_d} + \mathcal{D}_{bc_v} + \mathcal{D}_{bc_p}$ and $\gamma = 5/3$. $\mathcal{D}_{bc_d}, \mathcal{D}_{bc_v}, \mathcal{D}_{bc_p}$ are point sets of periodic boundary conditions.

### E.3 DIFFUSION REACTION EQUATION

The diffusion-reaction (DR) equation describes how concentration of a chemical spreads over time and space trough chemical reactions. In the two component case, with two latent functions $u = \psi(x, y, t)$ and $v = \phi(x, y, t)$ and Fitzhugh-Nagumo reaction equation, we have

$$\psi_t = D_u * \psi_{xx} + D_u * \psi_{yy} + \psi + \psi^3 - k - \phi \tag{24}$$

and

$$\phi_t = D_v * \phi_{xx} v + D_v * \phi_{yy} + \psi - \phi \tag{25}$$

where $D_u$ is the activator diffusion coefficient and $D_v$ is the inhibitor diffusion coefficient. In this dataset, the initial conditions $\psi(x, y, 0)$ and $\phi(x, y, 0)$ are normal distributions $\mathcal{N}(0, 1)$ boundary conditions are $D_u * \psi_x = 0$, $D_v * \phi_x = 0$, $D_u * \psi_y = 0$, and $D_v * \phi_y = 0$. Therefore, we use the

following loss function derived from these constraints in our experiment:

$$
\begin{aligned}
\mathcal{L}(\boldsymbol{\psi}; \mathcal{D}) = {} & \frac{1}{|\mathcal{D}_f|} \sum_{\mathbf{x} \in \mathcal{D}_f} \|f_1(\mathbf{x}) + f_2(\mathbf{x})\|_2^2 + \\
& \frac{1}{|\mathcal{D}_{ic_u}|} \sum_{\mathbf{x}, \mathbf{y} \in \mathcal{D}_{ic_u}} \|u(\mathbf{x}) - \mathbf{y}\|_2^2 + \frac{1}{|\mathcal{D}_{ic_v}|} \sum_{\mathbf{x}, \mathbf{y} \in \mathcal{D}_{ic_v}} \|v(\mathbf{x}) - \mathbf{y}\|_2^2 + \\
& \frac{1}{|\mathcal{D}_{bc}|} \sum_{\mathbf{x} \in \mathcal{D}_{bc}} \|u_x(\mathbf{x}) + v_x(\mathbf{x}) + u_y(\mathbf{x}) + u_y(\mathbf{x})\|_2^2 + \\
& \frac{1}{|\mathcal{D}_{bc_u}|} \sum_{\mathbf{x}, \mathbf{y} \in \mathcal{D}_{bc_u}} \|u(\mathbf{x}) - \mathbf{y}\|_2^2 + \frac{1}{|\mathcal{D}_{bc_v}|} \sum_{\mathbf{x}, \mathbf{y} \in \mathcal{D}_{bc_v}} \|v(\mathbf{x}) - \mathbf{y}\|_2^2
\end{aligned}
\tag{26}
$$

$$
f_1(\mathbf{x}) = u_t(\mathbf{x}) - D_u * u_{xx}(\mathbf{x}) - D_u * u_{yy}(\mathbf{x}) - u(\mathbf{x}) + u(\mathbf{x})^3 + k + v(\mathbf{x})
\tag{27}
$$

$$
f_2(\mathbf{x}) = v_t(\mathbf{x}) - D_v * v_{xx}(\mathbf{x}) - D_v * v_{yy}(\mathbf{x}) - u(\mathbf{x}) + v(\mathbf{x})
\tag{28}
$$

where $k = 0.005$, $D_u$ is the diffusion coefficient for activator and $D_v$ is the diffusion coefficient for inhibitor.

