# OpenReview forum: "ON TRAINING DERIVATIVE-CONSTRAINED NEURAL NETWORKS"
_ICLR.cc/2024/Conference — ICLR 2024 Conference Withdrawn Submission_

### Official Review · Reviewer_6cYD · 2023-10-31

**Soundness:** 3 good
**Presentation:** 3 good
**Contribution:** 3 good
**Rating:** 6
**Confidence:** 3

**Summary:**

This work proposes a series of technical improvements to better train neural networks with derivative constraints that are common in physics-informed settings. A new activation function, the integrated ReLU (IReLU) is proposed, along with other fixes such as denormalization and label rescaling.

**Strengths:**

- **The paper addresses an important open problem and presents concrete improvements over a comprehensive set of experiments.** Training derivative constrained networks is central to solving a lot of the physics-related applications, and it can be difficult in practice since it is hard to balance the derivative constraint in the total loss function. This paper proposes methods that address this difficulty by respecting the different numerical scales of the system. Improvements across many different network models as well as applications (quantum chemistry, fluid dynamics, diffusion-reaction, etc.) are shown in the experiment section.
- **Related work is discussed properly.** The authors adequately address the existing literature on derivative constrained neural networks and points out that their work differs from the literature in that they try to find general drop-in replacements that are suitable for a variety of different tasks and architectures.

**Weaknesses:**

- **The organization of the paper has room for improvement.** There is a motivation section in which the authors write about the experiments in detail, including the dataset size, numerical scales and units of the quantities of interest, but these details do not directly contribute to motivating the proposed methods.
- **Some observations are left unexplained/unexplored.** In section 3 results, the authors observed that the "energy loss divided by 1000 is typically much lower than the force loss," and that "it is not easy to improve the relative difference, even for large values of $\beta$." Why should this be the case? Why does the energy loss in figure 1(b) show high variance with high $\beta$? Despite this being mostly a methods paper, I wish there was more discussion providing insight and analysis on these empirical observations.
- **Some intuitions can be made more mathematically precise.** It is a good thing that the paper provides lots of intuition and insights on the problems and solutions presented. However, some of these intuitions can be difficult to understand for readers not familiar to the subject. For example in section 4.2 "...DC NNs are more sensitive to *units* compared to typical training without derivative constraints because of the linearity of derivatives", and "we can interpret the constant $c$ as determining the *units* of $x$". It is still unclear to me why some of the internal normalization (e.g. batch normalization) will not respect the units. It would be greatly helpful if the authors would include a short example of a concrete model and toy data to further elucidate this point.

**Questions:**

- On the results in section 3: why does it makes sense to divide the energy loss by 1000 in the first place? Would it be more reasonable to somehow plot the "variance explained by the model" instead of the unscaled losses?
- How is label rescaling different from the usual dataset-preprocessing? In your opinion, why didn't previous works consider this simple procedure?

---

> ### Author Response · Authors · 2023-11-22
> **Message from the authors (after withdrawal) posted by PC on authors' behalf; it is not required to respond**
>
> We thank the reviewer for their time and thoughtful feedback. We respond to the questions in turn.
>
> > On the results in section 3: why does it makes sense to divide the energy loss by 1000 in the first place? Would it be more reasonable to somehow plot the "variance explained by the model" instead of the unscaled losses?
>
> We thank the reviewer for this question. Dividing by 1000 is simply changing mili-eV to eV. This preserves the scale of the quantity since we are only changing the units we use. It would be interesting to try more complex measures, but our thought process was to begin with the simplest method first that does not change the scale.
>
> > How is label rescaling different from the usual dataset-preprocessing? In your opinion, why didn't previous works consider this simple procedure?
>
> We thank the reviewer for this question. We do not normalize our inputs and keep the scale of all physical quantities intact since it's useful for the network to know the distance between two atoms is twice as long as another pair since the interactions are not scale-invariant. Our experience training derivative-constrained networks is that they suffer instability and so our first instinct is to use a regularization method on the network.

---

### Official Review · Reviewer_LFn2 · 2023-11-01

**Soundness:** 2 fair
**Presentation:** 2 fair
**Contribution:** 2 fair
**Rating:** 3
**Confidence:** 4

**Summary:**

The paper tackles the problem of unstable training of derivative-constrained (DC) neural networks. Specifically, the authors proposed IReLU activation function to replace ReLU, which may lose training signals in DC settings. Furthermore, they also proposed eliminating normalization layers in current networks and rescaling the labels to reduce the sensitivity of the network with respect to units. Experiments were conducted on on a variety of architectures, datasets, and tasks including quantum chemistry and physics-informed neural networks.

**Strengths:**

* The paper tackles an important problem of stabilizing training of derivative-constrained neural networks.

* The writing of the paper is clear.

* Experiments are conducted on a wide range of tasks and architectures.

**Weaknesses:**

* The authors should introduce a name for their proposed method.

* The ideas of the papers are quite incremental and mostly based on intuition. For example, the authors hypothesize that derivative-constrained NNs are sensitive to units without any theoretical/empirical evidence.

**Questions:**

1. The experimental results are questionable, especially ones with quantum chemistry neural networks where we can observe huge improvements. Can the authors provide more detailed clarifications/explanations for these improvements?

2. Why label rescaling is not necessary for PINN experiments? Does this mean the label rescaling is only designed for quantum chemistry experiments? Again, the results on quantum chemistry datasets are questionable.

3. Have the authors tried alternatives of ReLU, such as SiLU, ELU, etc.?

4. I would appreciate if the authors included ablation studies to show the effectiveness of each component: iReLU, denormalization, label rescaling.

5. Normalization techniques were proposed to improve stability when training neural networks. Why removing them in the paper's setting can improve training stability? I would expect a more detailed answer other than the intuition of sensitivity to units mentioned in the paper.

6. Can IReLU be applied in derivative constraints involving higher-order derivatives, such as second-order? If not, can we have a more generic activation function for such cases?

I look forward to the authors' response and will be happy to increase my score if the authors can address my concerns.

---

> ### Author Response · Authors · 2023-11-22
> **Message from the authors (after withdrawal) posted by PC on authors' behalf; it is not required to respond**
>
> We thank the reviewer for their time and thoughtful feedback. We respond to the questions in turn.
>
> > The experimental results are questionable, especially ones with quantum chemistry neural networks where we can observe huge improvements. Can the authors provide more detailed clarifications/explanations for these improvements?
>
> We are also surprised to see huge improvements on some datasets. Our speculation is that for MD17, the absolute values of the energies are quite large compared to the forces, by several orders of magnitude, and that this makes it difficult to train without our methods. We still see improvements on OCP, but since the values of the energies and forces are closer, our improvements (when they occur), are less dramatic.
>
> > Why label rescaling is not necessary for PINN experiments? Does this mean the label rescaling is only designed for quantum chemistry experiments? Again, the results on quantum chemistry datasets are questionable.
>
> We thank the reviewer for this question. Label rescaling can be applied to PINN experiments as well. We will include further tests to confirm.
>
> > Have the authors tried alternatives of ReLU, such as SiLU, ELU, etc.?
>
> We thank the reviewer for this question. We have tried other integrated activation functions that are common in quantum chemistry networks, but decided to focus on RELU to keep the space of experiments tractable for a single paper, and to test this idea on a wider range of tasks and domains.
>
> > I would appreciate if the authors included ablation studies to show the effectiveness of each component: iReLU, denormalization, label rescaling.
>
> We thank the reviewer for this question. We included the ablation studies in the appendix and will clarify that we have ablation studies in the body of the paper.
>
> > Normalization techniques were proposed to improve stability when training neural networks. Why removing them in the paper's setting can improve training stability? I would expect a more detailed answer other than the intuition of sensitivity to units mentioned in the paper.
>
> We thank the reviewer for this question. Given the constraints of a single paper, we were not able to test out all the consequences of label rescaling on training stability beyond the intuition that if a multi-objective loss function has predictions that differ by orders-of-magnitude, rescaling the units of the label may help. We agree that it would be interesting to test the unit-sensitivity hypothesis more.
>
> > Can IReLU be applied in derivative constraints involving higher-order derivatives, such as second-order? If not, can we have a more generic activation function for such cases?
>
> We thank the reviewer for this question. Yes, as shown in our PINN experiments, we test on higher-order derivative cases as well. As we mention in the paper, it may be better to use other integrated activation functions as opposed to the integrated RELU. We have experimented with other activations, but have not tested them in the detail that we have done with the RELU.

---

### Official Review · Reviewer_WsvB · 2023-11-03

**Soundness:** 2 fair
**Presentation:** 1 poor
**Contribution:** 2 fair
**Rating:** 3
**Confidence:** 4

**Summary:**

Motivated by the convergence discrepancy of the loss terms in physics-informed training of neural networks, the paper proposes a new activation function, IRELU, and a label rescaling method, while abandoning common normalization techniques.

**Strengths:**

- The paper tries to address an important challenge with PINNs, concerning the discrepancy between loss terms in physics-informed training.
- The direction taken by authors in focusing on the units of derivatives and labels is interesting.

**Weaknesses:**

- As also pointed out by the authors, the proposed IRELU activation has limited usability in physics-informed models, where one might need derivatives of an arbitrary order w.r.t. inputs, while derivatives for IRELU are $0$ for third and higher order derivatives. Even for second order PDEs, the effects of a constant second derivative of $1$ in IRELU (for $x>0$) need more attention and study.
- Preventing vanishing and exploding gradients is one major characteristic of RELU. The gradient propagation of IRELU is not studied, though. As in your experiments, physics-informed training usually involves training with the solution data as well (IC, BC, etc.), where the first order derivative of IRELU ($=x$ for $x>0$) appears in the optimization. This is concerning for exploding gradients, especially since the paper also abandons normalization.
- The notion of the 'difficulty of learning derivative-constrained info based on the loss scaling term' is inaccurate. While convergence discrepancy of different loss terms is known to happen in physics-informed training, the convergence rate is shown to be in favor of the residual loss (derivative-constrained term) in some cases [1].
- Other works have studied activation functions in the physics-informed setting before [2, 3]. Lack of review and comparison with such works is surprising. Moreover, the authors do mention the adaptive loss scaling methods, but, there is again no comparison with those methods.


[1] Wang, S., Yu, X., & Perdikaris, P. (2020). When and why PINNs fail to train: A neural tangent kernel perspective. ArXiv. /abs/2007.14527

[2] Sitzmann, V., Martel, J., Bergman, A., Lindell, D., & Wetzstein, G. (2020). Implicit neural representations with periodic activation functions. Advances in neural information processing systems, 33, 7462-7473.

[3] Jagtap, A. D., Kawaguchi, K., & Karniadakis, G. E. (2020). Adaptive activation functions accelerate convergence in deep and physics-informed neural networks. Journal of Computational Physics, 404, 109136.

### Minor Comments
- There are a few grammatical errors, and the readability can also be improved.
- In Sec 3.2, Results, the references to Fig. 2 seem to be meant for Fig. 1.

**Questions:**

1. A more in-depth study of the proposed activation function would be really helpful. Authors may want to explain what characteristics IRELU shares with RELU and how it addresses the issues with polynomial activation functions.
2. The presentation and readability can be greatly improved by adding more plots instead of tables; Especially, in the experiments and ablation study to show how the proposed methods contribute to addressing the loss discrepancy. Also, the plotting style in Figures 1a and 1b is not informative and rather confusing.
3. Section 4.2 is very limited in justifying the proposed rescaling method and how it improves the learning of the derivative-constrained info. I would appreciate more details and insights regarding the choice of $C$ and why normalization methods are discouraged.
4. As mentioned in the Weaknesses, comparison with other activation functions that are designed for or tested with PINNs is crucial.

---

> ### Author Response · Authors · 2023-11-22
> **Message from the authors (after withdrawal) posted by PC on authors' behalf; it is not required to respond**
>
> We thank the reviewer for their time and thoughtful feedback. We respond to the questions in turn.
>
> >A more in-depth study of the proposed activation function would be really helpful. Authors may want to explain what characteristics IRELU shares with RELU and how it addresses the issues with polynomial activation functions.
>
> We thank the reviewer for this question. The derivative of an IRELU is a RELU. A polynomial activation function approximates a RELU with a polynomial so that it may have non-vanishing higher-order derivatives. However, this changes the behavior in that activations < 0 now can propagate signal, which a RELU and IRELU would block. We agree that a more in-depth study of the activation function would be helpful, and were limited by what we thought we could reasonably put in a single paper to explain how to train with a IRELU across a range of architectures and tasks for the case of derivative-constrained networks.
>
> > The presentation and readability can be greatly improved by adding more plots instead of tables; Especially, in the experiments and ablation study to show how the proposed methods contribute to addressing the loss discrepancy. Also, the plotting style in Figures 1a and 1b is not informative and rather confusing.
>
> The primary reason we choose tables is due to either order-of-magnitude differences in the performance across networks on the same task or the difference in units in measuring value prediction versus gradient/higher-order derivative prediction. We can revisit this to determine if there are better ways to plot the information.
>
> > Section 4.2 is very limited in justifying the proposed rescaling method and how it improves the learning of the derivative-constrained info. I would appreciate more details and insights regarding the choice of C and why normalization methods are discouraged.
>
> We thank the reviewer for this question. There are two reasons why we remove normalization methods. First, we observed when removing normalization methods improved the performance of networks. Second, physical quantities aren't scale invariant – 2 times as much mass requires 2 times as much force – so normalizing might destroy this information. The idea of C really is to predict in mm instead of m so that we can reduce the magnitudes of the numbers involved.
>
> > As mentioned in the Weaknesses, comparison with other activation functions that are designed for or tested with PINNs is crucial.
>
> We thank the reviewer for identifying PINN-specific activation functions and will test those activation functions on both PINNs as well as the quantum chemistry networks to see if there are performance differences.